# Optimal Decisions in a Multi-Party Closed-Loop Supply Chain Considering Green Marketing and Carbon Tax Policy

**DOI:** 10.3390/ijerph19159244

**Published:** 2022-07-28

**Authors:** Shan Lyu, Yuyu Chen, Lei Wang

**Affiliations:** 1School of Management, Qufu Normal University, Rizhao 276826, China; amber_lvs@163.com (S.L.); wanglei5804@163.com (L.W.); 2School of Business, Changshu Institute of Technology, Changshu 215500, China

**Keywords:** carbon tax policy, green marketing, closed-loop supply chain, retailer-dominated

## Abstract

Global warming and e-waste pollution are two major environmental pollution issues that have attracted widespread attention. The government has adopted various measures to reduce carbon emissions from businesses and to make manufacturers responsible for recycling e-waste. In the face of external pressures, more and more companies are implementing sustainable closed-loop supply chain (CLSC) management to reduce environmental pollution and achieve sustainable development. Therefore, it is essential to study the operational decisions of CLSC enterprises. This paper considers a sustainable CLSC consisting of two competing manufacturers and a dominant retailer. The government imposes a carbon tax on the retailer, and two manufacturers collect used products directly from their customers. We separately examine whether implementing green marketing by the retailer and the collaboration between the two manufacturers can improve their profits. By building decentralized CLSC mathematical models and applying game theory methods, we obtain that green marketing can increase profits for all CLSC members and improve return rates. The collaboration may yield higher total profits for two manufacturers than a decentralized solution, while the retailer’s profits may be lost under certain conditions. Finally, we perform several numerical analyses to find the relationship between unit carbon emission tax and social welfare and gain some managerial insights. The study gives key factors that CLSC companies should consider when making decisions to help them achieve sustainability and provides recommendations for the government to set a reasonable unit carbon tax.

## 1. Introduction

In recent years, waste electrical and electronic equipment (WEEE) has become the fastest-growing household waste stream in the world due to its high consumption rate, short life cycle, and low maintenance. According to the Global E-Waste Monitor 2020, the e-waste stream is expected to reach 74.7 million metric tons by 2030 [1]. The heavy metals and organic pollutants in WEEE are hazardous to the environment and human health. In order to alleviate environmental pressure and achieve sustainable development, governments worldwide have introduced stricter regulations to speed up companies’ implementation of WEEE collection [2,3]. Under external environmental pressure, many companies are starting to implement closed-loop supply chain (CLSC) management [4,5]. As an effective means of achieving combined economic and environmental benefits, CLSC includes forward and reverse supply chains managed in a coordinated manner to maximize the use of materials on hand while also preventing excessive remnants of hazardous material entering the environment [6,7,8]. For example, Apple avoids the equivalent of 2.6 million tonnes of mined rock by using recycled tin, gold, and tungsten in iPhone 13 models (https://www.apple.com/ie/environment/ accessed on 25 July 2022). Xerox collected more than 2.7 million pounds of product through its customer equipment return program in the United States during 2021 (https://www.xerox.com/en-us/about/ehs/reduce-waste accessed on 25 July 2022). For a complex multi-party system, it is vital to study the optimal decision-making of the participants [9,10].

With frequent extreme weather, rising sea levels, and mass species extinctions, global warming is attracting as much attention as WEEE. Governments are adopting various carbon emission reduction policies to reduce greenhouse gas emissions, among which carbon tax is widely considered one of the most effective market mechanisms to reduce carbon emissions [11,12]. By considering economic, environmental and social sustainability issues altogether, we have developed a sustainable CLSC framework [9,13,14]. Through a mathematical-modeling approach, Tao et al. [15] and Liu et al. [16] analyze that in a sustainable CLSC, as the price of carbon tax increases, the total cost increases, and profits shrink accordingly. Apart from external government subsidies [17], what a company itself can do to increase profits and sustainability has not been thoroughly examined. This paper proposes green marketing measures and examines the impact of green marketing on retailers’ profits and other members of sustainable CLSC.

The green marketing strategy aims to increase customers’ knowledge, awareness, and concern about environmental issues and to modify their purchase intentions and willingness to return end-of-use products as part of a corporate strategy to meet customer, stakeholder, organizational, and legal requirements [18,19]. To establish the image of an environmentally friendly company, Walmart has designed the Project Gigaton for suppliers to avoid gigatons of greenhouse gas emissions, a total of 416 million metric tons of emissions have been avoided through energy, waste, packaging, agriculture, forests, and product use and design since 2017 (https://corporate.walmart.com/planet/climate-change accessed on 25 July 2022). Under green marketing, Mondal and Giri [10] examine optimal recycler selection, Li et al. [20] focus on the impact of government subsidies on the supply chain, and Khan et al. [21] concentrate on green consumption behavior. There is no literature examining the collaboration of sustainable CLSC manufacturers when the downstream retailer has channel rights, is regulated by a carbon tax, and implements green marketing. As a follower of the channel, it is crucial to ensure the continued profitability of manufacturers. It is also significant to examine whether manufacturers can improve their profits by cooperating and what factors influence this decision.

This paper considers a decentralized sustainable CLSC consisting of two competing manufacturers and a dominant retailer, where the two manufacturers collect used products directly from their customers. Government imposes a carbon tax on the retailer to reduce carbon emissions. The retailer decides whether to adopt green marketing, and the manufacturer decides whether to cooperate. The equilibrium strategy, profit, and social welfare under different CLSC scenarios are investigated by using the Stackelberg and Nash game approach. In particular, we tried to address the following questions:

(1) Does a carbon tax mitigate environmental pollution and affect green marketing and social welfare? Is there an optimal carbon tax that maximizes social welfare? (2) Under the carbon tax policy, can implementing green marketing bring profit improvement to retailers? What is the impact of green marketing on other members of the CLSC? (3) Can collaboration bring profitability gains for competing manufacturers, and what factors influence the decision to collaborate?

The novelties of the article are summarized in two aspects: Firstly, in contrast to the common manufacturer–retailer CLSC model, we consider a sustainable CLSC consisting of two competing manufacturers and one dominant retailer, incorporate green marketing and carbon tax policy into the CLSC, and analyze the impact of these factors on decision making, which has not been addressed. Secondly, we consider environmental, economic, and social welfare factors in sustainable CLSCs and propose sustainable business strategies to companies.

The main contributions of this paper are as follows: Firstly, we give some advice on green marketing decisions for retailers under the carbon tax policy, and we also suggest operating tactics for retailers to cope when channel followers collaborate. Secondly, we examined the two manufacturers’ decisions on whether to cooperate under the retailer’s green marketing, the factors influencing these decisions, and the impact of these decisions on CLSC. Finally, the harm of carbon emissions to the social environment is included in social welfare, the impact of the carbon tax on green marketing and social welfare is studied, and suggestions for government carbon tax formulation are given.

The remainder of this paper is organized as follows. In Section 2, we briefly review the relevant literature and describe our differences. Section 3 introduces the problem and assumptions. In Section 4, we present three models and derive their equilibrium decisions and profits. In Section 5, we perform some comparative analyses of the optimal outcomes of three models. Section 6 is the numerical studies. The last section is our conclusions.

## 2. Literature Review

Our work is related to the following three streams of research: decisions in the supply chain under green marketing, decisions in the supply chain under carbon emission reduction policies, and the modeling approach applied to sustainable supply chains.

### 2.1. Decisions in the Supply Chain under Green Marketing

As consumers increasingly prefer environmentally friendly companies, more and more scholars are engaging in green marketing-related research. Hong and Guo [22] study several cooperation contracts within a green product supply chain and investigate their environmental performance. Li et al. [20] propose that in a two-echelon supply chain under a cap-and-trade scheme, both manufacturer and retailer tend to collaborate on green marketing when green technology is invested and subsidized. Khan et al. [21] reveal that green supply chain management and strategic green marketing orientation have positive and significant effects on green consumption intention. These studies only consider the impact of green marketing on the traditional forward channel, while it will be more complex when the supply chain includes the reverse channel. About the impact of green marketing on the CLSC, Esmaeili et al. [23] examine agents’ short- and long-term behavior in implementing the appropriate collecting strategy in a two-echelon CLSC and reveal the effect of green marketing elasticity on the long-term behavior. Mondal and Giri [10] present a two-period closed-loop green supply chain model with a single manufacturer and a single retailer to investigate the impact of green innovation, marketing effort, and collection rate of used products on supply chain decisions. Asghari et al. [24] examine a single-stage green CLSC in which the green manufacturer, retailer, and collector try to reform the environmental effects of their operations, products, and services across the value chain according to their environmental responsibilities.

Our work proposes a different model from previous studies in which the retailer is constrained by the carbon tax regulation and has the channel power. We compare the differences before and after the implementation of green marketing in this scenario and explore the impacts of green marketing on CLSC. We also analyzed the situation in which two manufacturers collaborated.

### 2.2. Decisions in the Supply Chain under Carbon Emission Reduction Policies

Facing the environmental problems caused by carbon emissions, more and more scholars are engaged in this field of research to develop a sustainable supply chain [25,26,27,28].

Moreover, the government has adopted economic incentives and constraints to curb companies’ carbon emissions, including carbon tax, cap-and-trade, and carbon emission capacity. Yu and Han [29] study the impact of the carbon tax on carbon emission and retail price and propose an innovative supply chain contract to effectively coordinate the supply chain, which integrates modified wholesale price or modified cost-sharing contract with a two-part tariff contract. Xu et al. [30] consider a supply chain consisting of a manufacturer and a retailer under cap-and-trade regulation, and coordinate it through wholesale price and cost-sharing contracts. Xu et al. [31] examine the coordination of a dual-channel supply chain with price discount contracts under mandatory carbon capacity regulation.

As for the more complex CLSC, Taleizadeh et al. [32] establish two collecting remanufacturing scenarios to study the impact of collecting–remanufacturing processes on carbon reduction, quality improvement, and CLSC performance. Mondal and Giri [33] investigate retailers’ competition and cooperation in a green CLSC consisting of one common manufacturer and two competing retailers under governmental intervention and cap-and-trade policy. Dou and Cao [17] measure the environmental and economic performances jointly of three CLSCs with different collectors under carbon tax regulation and find that the collection rate and the first-period product quantity have a piecewise monotonous relationship with tax price and emission intensities. Xing et al. [34] create a CLSC with risk-aversion characteristics, revealing that carbon emission trading price, consumers’ low-carbon awareness, and carbon emission are negatively correlated with the expected utility of manufacturer and retailer. Luo et al. [11] develop four game-theoretic models to evaluate the impact of carbon tax policy on manufacturing and remanufacturing decisions in a CLSC, finding that a carbon tax can effectively promote manufacturers to invest in carbon reduction technology or remanufacture to reduce carbon emissions. Wang et al. [35] investigate how legal constraints on the recycling rate of used products and carbon trading mechanisms affect the profits and other decisions of CLSC members. Zhou et al. [36] introduce green factors into the existing CLSC network and study the impact of carbon trading, green innovation efforts, and green consumers on the choice of remanufacturing strategies, and found that the choice of remanufacturing strategy was related to the carbon trading price.

In contrast to existing studies, we explored the implication between carbon tax policy and green marketing in a retailer-dominated CLSC. We solve for the optimal values of the carbon tax price to maximize social welfare.

### 2.3. Modeling Approaches Applied to Sustainable Supply Chains

Supply chain sustainability practices have emerged in the past decade, with many papers modeling sustainable CLSCs for different sales prices of new and remanufactured products [37,38,39]. Some papers developed sustainable supply chains in a two-period setting [40,41,42]. Our work focuses on the same selling price; consumers cannot identify the difference between them, and CLSC members make decisions in a single period [43]. In terms of return methods, single collection methods are more common [3,44], and some of the literature also considers multiple parties or channels for collection [45,46]. In addition, multiple game theoretic approaches are applied to the modeling process of sustainable supply chains. Wang et al. [44] build three two-tier game models, the Stackelberg–collusion model, the Stackelberg–Nash model, and the Stackelberg–Stackelberg model, to consider the potential collusive behavior of retailers and the interactive decisions of upstream manufacturers. Zheng et al. [47] employ cooperative and non-cooperative game theoretic analyses to characterize the interactions between CLSC parties and find an appropriate profit allocation scheme to coordinate the supply chain system with fairness concerns. Hosseini-Motlagh et al. [48] use channel coordination and Nash–Bertrand game theory approaches to analyze how competitive sales prices and collection rates are influenced by the remanufacturing and energy-saving efforts.

Distinct from the above studies, we use Stackelberg and Nash game theory to model the behavior modes of the CLSC parties. Additionally, we compared our study with previous studies, and the differences are detailed in Table 1.

## 3. Problem Description and Model Assumptions

### 3.1. Problem Description

Consider a decentralized CLSC consisting of two competing manufacturers and a dominant retailer, as shown in Figure 1. Specifically, two manufacturers collect used products directly from their customers through reverse flow and remanufacture them. In order to ensure a specific return rate and increase consumer motivation to return, two manufacturers make separate return investments and undertake collection costs. According to the WEEE Directive, there is no difference between products made from new raw materials and old products. Therefore, two manufacturers wholesale their remanufactured and new products to the retailer at the same price through forwarding flow. The retailer sells products to customers and generates carbon emissions. Facing the growing global warming problem, the government imposes a carbon tax on the channel’s leading retailer to reduce carbon emissions.

In the above scenario, this paper first examines whether the retailer can enhance its profitability through green marketing in the face of carbon tax pressure and the impact of green marketing on CLSC. We establish and comparatively analyze two decentralized systems for the retailer with (Model HG) and without (Model NG) green marketing. To improve the efficiency of the decentralized system when the retailer adopts green marketing, we further investigate whether two competing manufacturers can enhance their total profits by cooperating. Therefore, we perform a comparative analysis of the two models before (Model HG) and after (Model PC) manufacturer collaboration. Finally, to investigate whether the carbon tax policy is effective for the system we created and the impact of the implementation of the carbon tax policy on social welfare, we further investigate it through the social welfare function.

### 3.2. Model Assumptions

The following assumptions are made to ensure the reasonableness of the model and for further analysis.

**Assumption** **1.**
*CLSC members make decisions in a single period, and plenty of products on the market can be returned by the manufacturer [47]. Both manufacturers have established collection channels and have manufacturing and remanufacturing capabilities. The remanufactured goods do not fully satisfy market demand, and manufacturers still need to produce new products.*


**Assumption** **2.***We use *Δ* to indicate the unit cost savings from remanufacturing. To ensure that recycling remanufacturing activities are economically viable, we assume*$\Delta =c_{n}-c_{r}>b$. *The return rate of the manufacturer*i(i∈{1,2})*is formulated as*$\tau_{i}=\sqrt{I_{i}/C}$, $0<\tau_{i}<1$, *where*$I_{i}$*and C denote the investment in collection activities of the manufacturer i and scaling parameter; therefore, the total cost of collection is expressed as*$c\left(\tau_{i}\right)=b\tau_{i}d_{i}+C{\tau_{i}}^{2}$.

**Assumption** **3.**
*We use $E=e_{1}d_{1}+e_{2}d_{2}$ to denote the total carbon emissions. Define λE(λ>0) as the social loss of environmental pollution caused by the total carbon emission of the retailer, where λ represents the marginal social damage of carbon emission. The government sets the unit carbon emission tax $c_{t}$ on the retailer, the total carbon tax revenue is formulated as $\Pi_{G}=c_{t}E$. We use $\frac{1}{2}\eta s^{2}$ to model the retailer green marketing investment, where η and s denote the retailer green marketing investment cost coefficient and the green marketing level, respectively.*


**Assumption** **4.**
*A make-to-order (MTO) strategy is implemented in this system, and the market demand is given by $d_{i}=a_{i}-p_{i}+\alpha p_{3-i}+\beta s,i\in \left\{1,2\right\},$ where $a_{i}\left(>0\right)$ is the base market capacity of product i, α is the cross-price-sensitivity parameter and β measures the elasticity of demand regarding the green marketing level [24,51].*


**Assumption** **5.**
*To ensure that all models have the interior point solutions, we assume the scale parameter satisfies the following range $4C-\left(1+\alpha \right){\left(\Delta -b\right)}^{2}>0$ and $\eta \left(1-\alpha \right)\left[2C\left(2-\alpha \right)-\left(1-\alpha \right){\left(\Delta -b\right)}^{2}\right]-2\beta^{2}C>0$.*


In the following, we identify the retailer in the CLSC as “he”. Table 2 contains the main variables and parameters in the models.

## 4. Model Solution

### 4.1. Decentralized System without Green Marketing

Considering the CLSC mentioned above, we use the retailer-led Stackelberg–Bertrand game to model it. The game sequence is as follows: First, the government decides the unit carbon emission tax $c_{t}$ to maximize social welfare. Second, observing the unit carbon emission tax cost, the retailer decides the sale price $p_{i}$ to maximize his profit. Third, two manufacturers decide the wholesale prices $w_{i}$ and the return rate $\tau_{i}$ based on the decision of the retailer at the same time to maximize their respective profit. In the system without green marketing, the demand function is given by $d_{i}=a_{i}-p_{i}+\alpha p_{3-i},i\in \left\{1,2\right\}$.

The profit function of the retailer and two manufacturers are expressed as follows: (1)$$\Pi_{r}^{NG}\left(p_{1},p_{2}\right)=\left(p_{1}-w_{1}\right)d_{1}+\left(p_{2}-w_{2}\right)d_{2}-c_{t}\left(e_{1}d_{1}+e_{2}d_{2}\right),$$
(2)$$\Pi_{m_{i}}^{NG}\left(w_{i},\tau_{i}\right)=w_{i}d_{i}-\left[\left(1-\tau_{i}\right)c_{n}+\tau_{i}c_{r}\right]d_{i}-\left[b\tau_{i}d_{i}+C\tau_{i}^{2}\right],i=1,2.$$

In Equation (Equation 1), the first two parts are the revenue earned by the retailer from selling two different products, and the last part is the total carbon taxes. In Equation (Equation 2), the first part is the revenue earned by the manufacturer from selling the product to the retailer, the second part is the total production cost, and the third part is the total cost of collection.

According to the relevant literature, the generalized social welfare function consists of two components, a positive economic utility, and a disutility term [52,53,54]. Similar to the model of Moraga-Gonzalez and Padron-Fumero [52], the social welfare function is given by
(3)$$W=\Pi_{m}+\Pi_{r}+\Pi_{G}+C_{s}-\lambda E.$$

The first four terms represent positive economic utility, which are manufacturers’ and retailer’s total profit, the government’s carbon tax revenue, and consumer surplus. The last component represents the social loss of environmental pollution caused by the total carbon emissions of the retailer.

Following Johari and Hosseini-Motlagh [55], the consumer surplus function is expressed as $C_{s}=\int_{p_{1min}}^{p_{1max}}d_{1}dp_{1}+\int_{p_{2min}}^{p_{2max}}d_{2}dp_{2}$, where $p_{imin}$ denote the market price of product *i* and $p_{imax}$ denotes the maximizer price that the consumers are willing to pay for the product i(i∈{1,2}).

We apply the backward induction method to derive the equilibrium solutions. The main results are summarized in Table 3.

**Proof** See Appendix A. □

### 4.2. Decentralized System with Green Marketing

In this section, the game sequence is as follows: First, the government decides the unit carbon emission tax $c_{t}$ to maximize social welfare. Second, observing the unit carbon emission tax cost, the retailer decides the sale price $p_{i}$ and green marketing level *s* to maximize his profit. Third, two manufacturers decide the wholesale prices $w_{i}$ and the return rate $\tau_{i}$ based on the decision of the retailer at the same time to maximize their respective profit. In the system with green marketing, the demand function is given by $d_{i}=a_{i}-p_{i}+\alpha p_{3-i}+\beta s,i\in \left\{1,2\right\}$.

The profit functions of the retailer and two manufacturers are expressed as follows: (4)$$\Pi_{r}^{HG}\left(p_{1},p_{2},s\right)=\left(p_{1}-w_{1}\right)d_{1}+\left(p_{2}-w_{2}\right)d_{2}-\frac{1}{2}\eta s^{2}-c_{t}\left(e_{1}d_{1}+e_{2}d_{2}\right),$$
(5)$$\Pi_{m_{i}}^{HG}\left(w_{i},\tau_{i}\right)=w_{i}d_{i}-\left[\left(1-\tau_{i}\right)c_{n}+\tau_{i}c_{r}\right]d_{i}-\left[b\tau_{i}d_{i}+C\tau_{i}^{2}\right],i=1,2.$$

We apply the backward induction method to derive the equilibrium solutions. The main results are summarized in Table 4.

**Proof** See Appendix B. □

### 4.3. Partial Cooperation System with Green Marketing

In this section, we model the case of partial cooperation, where two manufacturers form a coalition. The decision sequence is roughly the same as the HG model: First, the government decides the unit carbon emission tax $c_{t}$ to maximize social welfare. Second, observing the unit carbon emission tax cost, the retailer decides the sale price $p_{i}$ and green marketing level *s* to maximize his profit. Third, two manufacturers jointly decide the wholesale prices $w_{i}$ and the return rate $\tau_{i}$ based on the retailer’s decision to maximize their profit.

The profit of the retailer and the manufacturer are expressed as
(6)$$\Pi_{r}^{PC}\left(p_{1},p_{2},s\right)=\left(p_{1}-w_{1}\right)d_{1}+\left(p_{2}-w_{2}\right)d_{2}-\frac{1}{2}\eta s^{2}-c_{t}\left(e_{1}d_{1}+e_{2}d_{2}\right),$$
and
(7)$$\begin{array}{cc} \Pi_{m}^{PC}\left(w_{1},w_{2},\tau_{1},\tau_{2}\right) & =w_{1}d_{1}-\left[\left(1-\tau_{1}\right)c_{n}+\tau_{1}c_{r}\right]d_{1}-\left[b\tau_{1}d_{1}+C\tau_{1}^{2}\right]\\ \; & +w_{2}d_{2}-\left[\left(1-\tau_{2}\right)c_{n}+\tau_{2}c_{r}\right]d_{2}-\left[b\tau_{2}d_{2}+C\tau_{2}^{2}\right], \end{array}$$
respectively.

We apply the backward induction method to derive the equilibrium solutions. The main results are summarized in Table 5.

**Proof** See Appendix C. □

## 5. Model Analysis

We perform comparative analysis of the equilibrium decisions and profits of the three models in Section 4 and obtain the following propositions.

**Proposition** **1.**
*Comparing the scenarios of retailer with and without green marketing, for i=1,2, the following hold:*
(1)

$\Pi_{r}^{HG*}>\Pi_{r}^{NG*},\Pi_{m_{i}}^{HG*}>\Pi_{m_{i}}^{NG*}.$

(2)

$\tau_{i}^{HG*}>\tau_{i}^{NG*},E^{HG*}>E^{NG*}.$




**Proof** See Appendix D. □

Proposition 1 indicates that the retailer gains more profit when he engages in green marketing. The retailer’s green marketing investment boosts its profits and those of other CLSC members. Green marketing can increase demand by engaging environmentally conscious consumers, thus increasing manufacturer return rate and total carbon emissions.

**Proposition** **2.**
*In the case of retailer’s participation in green marketing, comparing manufacturers’ cooperation and non-cooperation, the following hold:*
(1)
*$s^{HG}=\Omega s^{PC}$, where $\Omega =\frac{\left(1-\alpha \right)\eta \left[4C-\left(1-\alpha \right){\left(\Delta -b\right)}^{2}\right]-2\beta^{2}C}{\left(1-\alpha \right)\eta \left[2C\left(2-\alpha \right)-\left(1-\alpha \right){\left(\Delta -b\right)}^{2}\right]-2\beta^{2}C}>1$.*
(2)
*When $\Omega_{1}\leq \theta_{2}$, $\Pi_{m}^{PC}\geq \Pi_{m}^{HG}$, otherwise, $\Pi_{m}^{PC}<\Pi_{m}^{HG}$,*

*where*

$\Omega_{1}=\left(1-\alpha \right)\eta \left[2\beta^{2}-\alpha \left(1-\alpha \right)\eta \right]\left[4C-\left(1-\alpha \right){\left(\Delta -b\right)}^{2}\right]-4\beta^{4}C,$



$\theta_{2}=\frac{\beta^{2}\alpha C^{2}{\left[a_{1}-a_{2}-\left(1+\alpha \right)c_{t}\left(e_{1}-e_{2}\right)\right]}^{2}{\left\{\left(1-\alpha \right)\eta \left[4C-\left(1-\alpha \right){\left(\Delta -b\right)}^{2}\right]-2\beta^{2}C\right\}}^{2}}{\eta \left(1-\alpha^{2}\right){s^{PC}}^{2}\left[4C-\left(1+\alpha \right){\left(\Delta -b\right)}^{2}\right]\left[2C\left(2+\alpha \right)-\left(1+\alpha \right){\left(\Delta -b\right)}^{2}\right]}.$

(3)*When*$\Omega -1\leq \theta_{1}$, $\Pi_{r}^{PC}\geq \Pi_{r}^{HG}$, otherwise, $\Pi_{r}^{PC}<\Pi_{r}^{HG}$,*where*$\theta_{1}=\frac{2\beta \alpha C^{2}{\left[a_{1}-a_{2}-\left(1+\alpha \right)c_{t}\left(e_{1}-e_{2}\right)\right]}^{2}}{\eta \left(1+\alpha \right)\left[4C-\left(1+\alpha \right){\left(\Delta -b\right)}^{2}\right]\left[2C\left(2+\alpha \right)-\left(1+\alpha \right){\left(\Delta -b\right)}^{2}\right]\left[a_{1}+a_{2}-\left(1-\alpha \right)\left(2c_{n}+c_{t}e_{1}+c_{t}e_{2}\right)\right]s^{PC}}$.(4)*When*$\Omega -1\leq \theta_{2}$, $E^{PC}\geq E^{HG}$, otherwise, $E^{PC}<E^{HG}$,*where*$\theta_{2}=\frac{2\beta \alpha C^{2}\left(e_{1}-e_{2}\right)\left[a_{1}-a_{2}-\left(1+\alpha \right)c_{t}\left(e_{1}-e_{2}\right)\right]}{\left(e_{1}+e_{2}\right)\left(1-\alpha \right)\eta s^{PC}\left[4C-\left(1+\alpha \right){\left(\Delta -b\right)}^{2}\right]\left[2C\left(2+\alpha \right)-\left(1+\alpha \right){\left(\Delta -b\right)}^{2}\right]}$.


**Proof** See Appendix E. □

From Proposition 2 (1)–(2), we find that retailer’s green marketing level is lower in the partial cooperation scenario than in the decentralized scenario. Manufacturers’ total profits may be higher in the partial cooperation scenario than in the decentralized scenario under certain conditions. Proposition 2 (3)–(4) provide thresholds to compare the difference in total carbon emission and the profit of the retailer between the two models, and the thresholds are related to the retailer’s green marketing level. Under certain conditions, the retailer’s profits may be lost.

**Proposition** **3.**
*In Model y, where y∈{NG,HG,PC}, the following hold:*
(1)
*$\frac{\partial E^{NG*}}{\partial c_{t}}<0$, $\frac{\partial E^{HG*}}{\partial c_{t}}<0$, $\frac{\partial E^{PC*}}{\partial c_{t}}<0$.*
(2)
*$\frac{\partial s^{HG*}}{\partial c_{t}}<0$, $\frac{\partial s^{PC*}}{\partial c_{t}}<0$.*



**Proof** See Appendix F. □

Proposition 3 indicates that total carbon emissions decrease with the increase of carbon tax in all models. The level of green marketing by the retailer will decrease with carbon tax increase.

## 6. Numerical Analysis

In this section, we deliver several numerical examples to illuminate the above theoretical results and gain managerial insights. With reference to the previous parameter values [51,56], we assume that $a_{1}=180,\;a_{2}=100,\;\alpha =0.3,\;C=1000,\;c_{n}=20,\;c_{r}=5,\;\Delta =15,\;b=5,\;$$\eta =24,\;\beta =1.5,e_{1}=10,\;e_{2}=8.$ Plotting the social welfare function in the decentralized model with green marketing, we obtain Figure 2, which depicts the effects of $c_{t}$ and λ on $W^{HG}$. By analyzing Figure 2, we obtain that there exists an optimal value of positive $c_{t}$ to maximize *W* when λ is higher than 13.69. Otherwise, social welfare decreases as $c_{t}$ increases. In the numerical example below, we will randomly take λ=10 and $c_{t}=3$.

The equilibrium solutions of the three models are summarized in Table 6, and from the calculations in Table 6, we have the following sorts of observations:(1)The selling price, wholesale price, and return rate of both products, social welfare, and the profit of both manufacturers and retailer in the HG model are higher than those in the NG model. As a downstream firm, the retailer takes green marketing, but the benefits are less than those of upstream firms. Specifically, after the retailer engages in green marketing, the profits of manufacturers 1 and 2 improve by 14.47% and 24.73%, respectively, while the profits of the retailer increase by 8.37%. As the channel leader, the retailer decides the level of green marketing, takes green marketing measures, and raises the selling price of both products for less investment pressure and more profit. Manufacturers are incentivized by the retailer’s green marketing to increase return rates through improving their collection activity investment and to avoid profit decline through higher wholesale prices. Hence, the retailer’s green marketing not only increases his profits but also increases the social welfare and profits of both manufacturers.(2)In the PC model, the selling price, wholesale price, and total profits of manufacturers are higher than those in the HG model. The return rate, green marketing level, the profits of the retailer, and social welfare are lower than those in the HG model. Although manufacturers’ profits are higher than before the cooperation, the trend in total profits is down, as the retailer’s profits have fallen much more than the manufacturers’ growth. Under the manufacturers cooperative model, demand for both products, total profits, government carbon tax revenues, and consumer surplus are lower than before the cooperation. Despite lower environmental pollution, total social welfare is also lower than before the cooperation.

Figure 3 depicts the effects of $c_{t}$ on social welfare and total carbon emissions, and from the calculations in Figure 3, we have the following sorts of observations:(1)Figure 3a shows that social welfare decreases in all three models as the unit carbon tax increases when λ=10. The value of social welfare is influenced by the interaction of positive and negative economic utility. As $c_{t}$ increases, the social welfare value of the HG model is always greater than that of the PC model, while the social welfare values of the NG and PC models gradually approach.(2)Figure 3b shows that there exists an optimal value of $c_{t}$ to maximize *W* in all three models when λ=15, which are $c_{t}^{NG}=6.29$, $c_{t}^{HG}=4.07$, and $c_{t}^{PC}=0.77$. In different models, the optimal value of $c_{t}$ is influenced by the marginal social damage of carbon emissions and there exists a threshold value about λ. When λ exceeds the threshold, there exists an optimal value of $c_{t}$ to maximize social welfare; otherwise, social welfare will decrease as $c_{t}$ increases.(3)Figure 3c shows that in all three models, total carbon emission decreases as the unit carbon tax increase when λ=10. Total carbon emissions are the highest in the HG model, followed by the NG model and the lowest in the PC model. The government’s carbon tax on the retailer will reduce total carbon emissions, mitigating environmental pollution. Comparing Figure 3a, we can obtain a corresponding reduction in the total value of social welfare.

Figure 4 depicts the effects of Δ on profit, and from the calculations in Figure 4, we have the following sorts of observations:(1)Figure 4a shows that in the NG and HG models, the profits of both manufacturers increase as the unit cost savings from remanufacturing Δ increase, and the profits of both manufacturers in the HG model are higher than those in the NG model. Consistent with Proposition 1, green marketing inputs by the retailer can lead to better profits for manufacturers. As unit cost savings from remanufacturing increase, the manufacturer’s total cost of production decreases, and the manufacturer is encouraged to increase recycling investments to improve recovery rates, thereby increasing profits.(2)Figure 4b shows that in the HG and PC models, the total profits of the manufacturers increase as the unit cost savings from remanufacturing Δ increase, and the total profits of both manufacturers in the PC model are higher than that in the HG model.(3)Figure 4c shows that in all three models, the profits of the retailer increase as the unit cost savings from remanufacturing Δ increase, and the retailer’s profit is highest in the HG model, followed by the NG model, and lowest in the PC model.(4)Figure 4d shows that in all three models, the selling price of both products decreases as the unit cost savings from remanufacturing Δ increase, and the selling price is highest in the PC model, followed by the HG model, and lowest in the NG model. Since Product 1 has a higher base market capacity, its prices are higher than those of Product 2. When manufacturers’ production costs are low, the retailer will stimulate consumption and increase demand by lowering the selling price.

Figure 5 depicts the effects of β on profit, and from the calculations in Figure 5, we have the following sorts of observations:(1)Figure 5a shows that in the HG model, the profits of both manufacturers increase as the green marketing elasticity parameter of demand β increases, and the profits of both manufacturers in the HG model are higher than those in the NG model. Consistent with Proposition 1, green marketing inputs by the retailer can lead to better profits for manufacturers. As the green marketing elasticity parameter of the demand β increases, the gap between the two models increases.(2)Figure 5b shows that in the HG and PC models, the total profits of both manufacturers increase as the green marketing elasticity parameter of the demand β increases. There exists a threshold value of β; when β is lower than 1.68, the total profits of the manufacturers in the PC model are higher than the HG model. Consistent with Proposition 2, cooperation does not always lead to an increase in the total profits of manufacturers.(3)Figure 5c shows that retailer’s profits increase with the green marketing elasticity parameter of demand β in the HG and PC models. The retailer’s profits are higher in the HG model than in the NG model, the retailer’s profits are lowest in the PC model at the beginning, and as β increases, they will surpass the NG model.(4)Figure 5d shows that the selling price of both products increases as the green marketing elasticity parameter of demand β increases in the HG and PC models, and the selling price is highest in the PC model, followed by the HG model, and is lowest in the NG model. β can also be interpreted as the environmentally friendly preference of the consumers. Through green marketing, the consumers’ green consumption habits are continuously awakened and formed, leading to a certain product price tolerance. Therefore the retailer can increase the product price after engaging in green marketing.

## 7. Conclusions and Future Research Opportunities

### 7.1. Conclusions

This paper proposes a CLSC consisting of two competing manufacturers and a dominant retailer. The government imposes a carbon tax on the retailer, and two manufacturers collect used products directly from their customers. We first compare two models with and without the retailer considering green marketing; after which, we compare two scenarios of manufacturer cooperation and non-cooperation in green marketing. We analyze the equilibrium strategies and profits of each CLSC member, and consider the impact of the carbon tax on CLSC and social welfare. We finally set up some numerical analyses to test the theoretical conclusions and obtain the results.
(1)The carbon tax policy can be used to reduce corporate carbon emissions, which will decrease with the increase of carbon tax per unit. From the perspective of social welfare, an increase in the carbon tax price will lead to a decrease in social welfare under certain conditions. When the marginal social damage of carbon emissions is relatively low, social welfare will decrease as carbon taxes increase, while there exists an optimal value of carbon tax that maximizes social welfare when the marginal social damage of carbon emissions is relatively high.(2)Green marketing is an effective means of improving the social welfare and profitability of all CLSC members. The implementation can attract environmentally friendly consumers, raise awareness of environmental issues among other consumers, change their consumption habits and increase their willingness to recycle products. Therefore, after the retailer implements green marketing, there is a corresponding increase in manufacturers’ return rate. This can reduce environmental pollution from e-waste and carbon emissions from the extraction and refinement of raw materials. The level of green marketing will be affected by the carbon tax policy. As the unit carbon tax increases for the HG and PC models, the optimal level of green marketing will decrease. That is, when the retailer faces higher carbon tax pressure, he will choose to reduce the level of green marketing appropriately to reduce the total cost. The retailer engaging in green marketing leads to an increase in demand and thus carbon emissions, but it is beneficial from a social welfare perspective.(3)Cooperation can increase manufacturers’ voice in CLSC, and it can increase manufacturers’ total profit under certain conditions. The value of consumer preference for environmental friendliness determines whether manufacturers are more profitable after the partnership than before. When the value of consumers’ preference for environmental friendliness is relatively high, the total profit after cooperation may be lower than before cooperation. The manufacturer’s cooperation may damage the retailer’s profits, and he will reduce green marketing investment or increase prices to ensure profits.

### 7.2. Managerial Implications

Based on the analysis results of the model, we provide the following management insights to the government and CLSC enterprises.

For the government, in the face of the growing global warming problem, implementing a carbon tax policy can effectively reduce the environmental pollution caused by carbon emissions. However, the implementation of carbon tax policies may bring about a decrease in total social welfare. The marginal social damage of carbon emissions determines the impact of the unit carbon tax on social welfare. Therefore, when the government sets the unit carbon tax value, they should refer to the marginal social damage of carbon emissions and pay attention to the changes in social welfare.

For the dominant retailer, the implementation of green marketing can effectively increase his profits under the pressure of a carbon tax. Green marketers can brand themselves as green and awaken the green awareness of consumers. The retailer can increase the selling price after implementing green marketing. The selling price setting should also consider the values such as consumer preferences for environmental friendliness and savings in remanufacturing costs. When the unit carbon tax increases, the retailer should reduce the level of green marketing to reduce cost pressure. The retailer should also reduce the level of green marketing when the two manufacturers collaborate.

For the manufacturers, they should actively participate in the green marketing activities of retailers, as green marketing can increase sales and therefore their profits. Manufacturers should refer to the value of consumers’ preference for environmental friendliness to decide whether to cooperate or not.

### 7.3. Limitations and Future Research Opportunities

The main limitations of this paper and further research opportunities are summarized below. First, we assume deterministic demand in this CLSC and use artificial data for numerical analysis, which differs slightly from the actual market. Considering uncertain demand may yield some other insights, and using data from case studies would be more representative. Second, this may have other implications when governments tax manufacturers’ carbon emissions and manufacturers invest in emissions-reducing technologies. Finally, we may change the model in different power structures and collectors to obtain the optimal equilibrium results.

## Figures and Tables

**Figure 1 ijerph-19-09244-f001:**
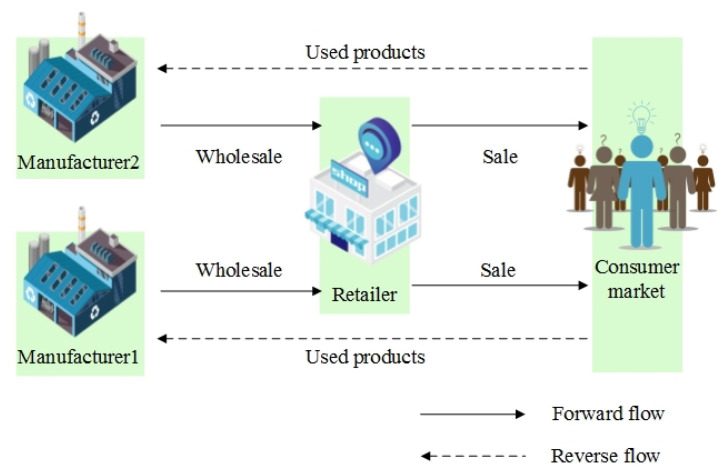
Decentralized model with manufacturer remanufacturing.

**Figure 2 ijerph-19-09244-f002:**
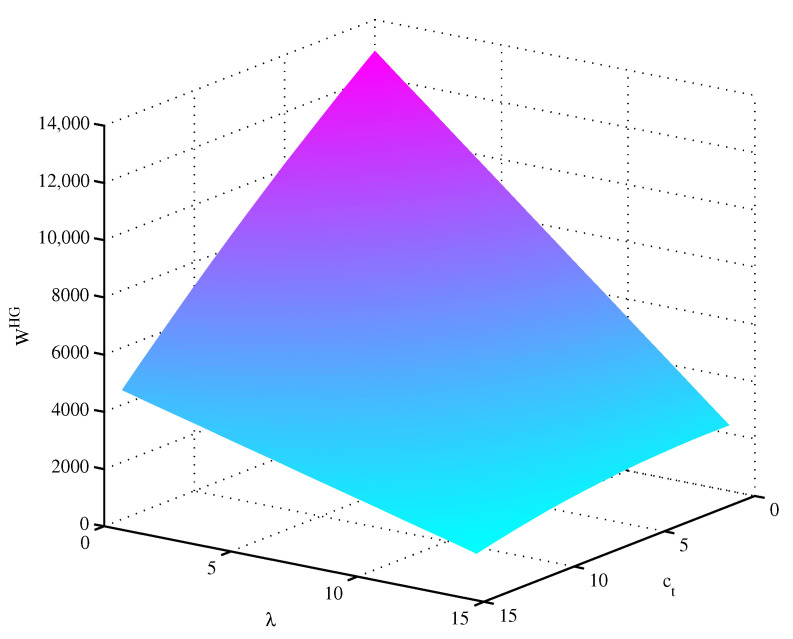
Effects of $c_{t}$ and λ on $W^{HG}$.

**Figure 3 ijerph-19-09244-f003:**
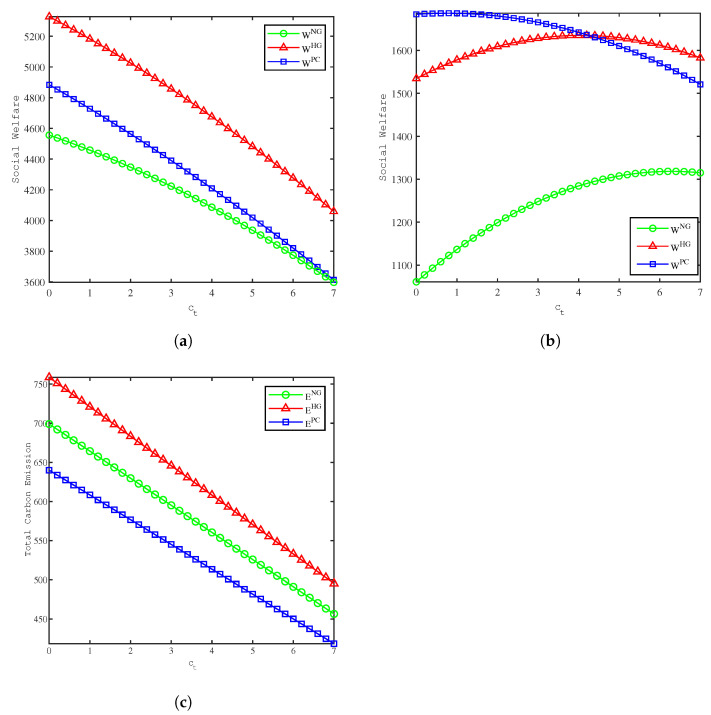
Comparison of the social welfare and total carbon emission between models. (**a**) Effects of $c_{t}$ on social welfare when λ=10. (**b**) Effects of $c_{t}$ on social welfare when λ=15. (**c**) Effects of $c_{t}$ on total carbon emission.

**Figure 4 ijerph-19-09244-f004:**
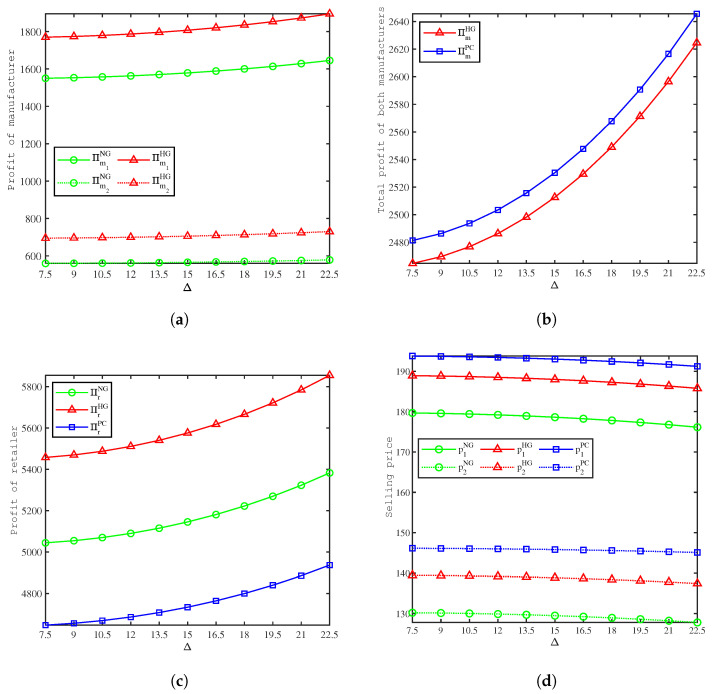
Comparison of the profit between models. (**a**) Effects of Δ on the manufacturers’ profits. (**b**) Effects of Δ on the manufacturers’ total profits. (**c**) Effects of Δ on the retailer’s profits. (**d**) Effects of Δ on the selling price.

**Figure 5 ijerph-19-09244-f005:**
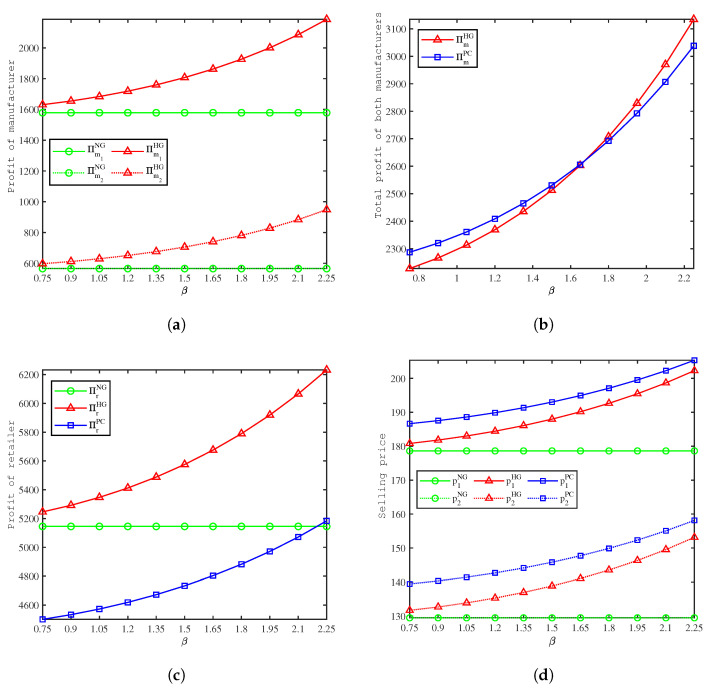
Comparison of the profit between models. (**a**) Effects of β on the manufacturers’ profits. (**b**) Effects of β on the manufacturers’ total profits. (**c**) Effects of β on the retailer’s profits. (**d**) Effects of β on the selling price.

**Table 1 ijerph-19-09244-t001:** The main differences between our works and previous studies.

References	Game Theory Approach	Coalition Structure	Carbon Emission Reduction Policy	Social Welfare	Green Marketing
Mondal and Giri [10]	S				*√*
Dou and Cao [17]	S		CTP		
Li et al. [20]	S		CTR		*√*
Hong and Guo [22]	S			*√*	*√*
Asghari et al. [24]	C	*√*			*√*
Yu and Han [29]	S		CTP		
Xu et al. [31]	S		CEC		
Xing et al. [34]	S		CTM		
Wang et al. [35]	S		CTM	*√*	
Zhang et al. [37]	S			*√*	
Wang et al. [44]	S, N	*√*		*√*	
Zheng et al. [47]	C, S	*√*			
Xue and Sun [49]	S, N	*√*	CTR	*√*	
Hosseini-Motlagh et al. [48]	N	*√*			
Zhou et al. [50]	S		CTP	*√*	
This paper	S, N	*√*	CTP	*√*	*√*

Note: √: Covered; S: Stackelberg Game; C: Cooperative Game; N: Nash Game; CTP: Carbon Tax Policy; CTR:
Cap-and-Trade Regulation; CEC: Carbon Emission Capacity Regulation; CTM: Carbon Trading Mechanisms.

**Table 2 ijerph-19-09244-t002:** Model parameters and decision variables.

Symbol	Definition
Parameters	
$d_{i}$	Demand of product *i*, i=1,2
$a_{i}$	Base market capacity of product *i*, i=1,2
α	Cross-price-sensitivity parameter of the demand function, 0<α<1
*C*	Scaling parameter between collecting investment and return rate
$c_{n}$	Unit cost of producing a new product
$c_{r}$	Unit cost of producing a remanufactured product, $c_{r}<c_{n}$
Δ	Unit cost savings from remanufacturing, $\Delta =c_{n}-c_{r}>0$
*b*	Transfer price of unit used product from consumers to the manufacturer
β	Green marketing elasticity parameter of the demand, β>0
η	Coefficient of the retailer’s green marketing investment, η>0
$e_{i}$	Unit carbon emission of product *i*, i=1,2
*E*	Total carbon emission
λ	Marginal social damage of carbon emission, λ>0
$\Pi_{x}^{y}$	Total profit of firm *x* in model *y*, where $x\in \{m_{1},m_{2},m,r\}$ and y∈{NG,HG,PC}
*W*	Social welfare
Decision variables	
$w_{i}$	Unit wholesale price of the product *i*, i=1,2
$\tau_{i}$	Return rate of the manufacturer *i*, i=1,2
*s*	Green marketing level of the retailer
$p_{i}$	Unit selling price of the product *i*, i=1,2
$c_{t}$	Unit carbon emission tax

**Table 3 ijerph-19-09244-t003:** Main results for the decentralized model of retailer without green marketing, *i* = 1, 2.

Notations	Values
$p_{i}^{NG*}$	$\frac{C\left(1-\alpha^{2}\right)\left[a_{3-i}-c_{t}\left(e_{3-i}-e_{i}\right)\right]+\left(a_{i}+\alpha a_{3-i}\right)\left[7C+5\alpha C-2\left(1+\alpha \right){\left(\Delta -b\right)}^{2}\right]}{2\left(1-\alpha^{2}\right)\left[2C\left(2+\alpha \right)-\left(1+\alpha \right){\left(\Delta -b\right)}^{2}\right]}-\frac{C[a_{1}+a_{2}-\left(1-\alpha \right)\left(2c_{n}+c_{t}e_{1}+c_{t}e_{2}\right)]}{2\left(1-\alpha \right)[2C\left(2-\alpha \right)-\left(1-\alpha \right){\left(\Delta -b\right)}^{2}]}$
$w_{i}^{NG*}$	$\frac{8c_{n}C+\left[2C-{\left(\Delta -b\right)}^{2}\right]\left[a_{i}-a_{3-i}+\left(1+\alpha \right)\left(4c_{n}-c_{t}e_{i}+c_{t}e_{3-i}\right)\right]}{4[2C\left(2+\alpha \right)-\left(1+\alpha \right){\left(\Delta -b\right)}^{2}]}+\frac{\left[2C-{\left(\Delta -b\right)}^{2}\right]\left[a_{1}+a_{2}-\left(1-\alpha \right)\left(2c_{n}+c_{t}e_{1}+c_{t}e_{2}\right)\right]}{4[2C\left(2-\alpha \right)-\left(1-\alpha \right){\left(\Delta -b\right)}^{2}]}$
$\tau_{i}^{NG*}$	$\frac{\left(\Delta -b\right)[a_{i}-a_{3-i}-\left(1+\alpha \right)c_{t}\left(e_{i}-e_{3-i}\right)]}{4[2C\left(2+\alpha \right)-\left(1+\alpha \right){\left(\Delta -b\right)}^{2}]}+\frac{\left(\Delta -b\right)[a_{1}+a_{2}-\left(1-\alpha \right)\left(2c_{n}+c_{t}e_{1}+c_{t}e_{2}\right)]}{4[2C\left(2-\alpha \right)-\left(1-\alpha \right){\left(\Delta -b\right)}^{2}]}$
$\Pi_{m_{i}}^{NG*}$	$\frac{4C-{\left(\Delta -b\right)}^{2}}{4C}{\left(\frac{C[a_{i}-a_{3-i}-\left(1+\alpha \right)c_{t}\left(e_{i}-e_{3-i}\right)]}{2[2C\left(2+\alpha \right)-\left(1+\alpha \right){\left(\Delta -b\right)}^{2}]}+\frac{C[a_{1}+a_{2}-\left(1-\alpha \right)\left(2c_{n}+c_{t}e_{1}+c_{t}e_{2}\right)]}{2[2C\left(2-\alpha \right)-\left(1-\alpha \right){\left(\Delta -b\right)}^{2}]}\right)}^{2}$
$\Pi_{r}^{NG*}$	$\frac{C{\left[a_{1}-a_{2}-\left(1+\alpha \right)c_{t}\left(e_{1}-e_{2}\right)\right]}^{2}}{4\left(1+\alpha \right)[2C\left(2+\alpha \right)-\left(1+\alpha \right){\left(\Delta -b\right)}^{2}]}+\frac{C{\left[a_{1}+a_{2}-\left(1-\alpha \right)\left(2c_{n}+c_{t}e_{1}+c_{t}e_{2}\right)\right]}^{2}}{4\left(1-\alpha \right)[2C\left(2-\alpha \right)-\left(1-\alpha \right){\left(\Delta -b\right)}^{2}]}$
$E^{NG*}$	$\frac{\left(e_{1}-e_{2}\right)C\left[a_{1}-a_{2}-\left(1+\alpha \right)c_{t}\left(e_{1}-e_{2}\right)\right]}{2[2C\left(2+\alpha \right)-\left(1+\alpha \right){\left(\Delta -b\right)}^{2}]}+\frac{\left(e_{1}+e_{2}\right)C\left[a_{1}+a_{2}-\left(1-\alpha \right)\left(2c_{n}+c_{t}e_{1}+c_{t}e_{2}\right)\right]}{2[2C\left(2-\alpha \right)-\left(1-\alpha \right){\left(\Delta -b\right)}^{2}]}$
$\Pi_{G}^{NG*}$	$\frac{c_{t}\left(e_{1}-e_{2}\right)C\left[a_{1}-a_{2}-\left(1+\alpha \right)c_{t}\left(e_{1}-e_{2}\right)\right]}{2[2C\left(2+\alpha \right)-\left(1+\alpha \right){\left(\Delta -b\right)}^{2}]}+\frac{c_{t}\left(e_{1}+e_{2}\right)C\left[a_{1}+a_{2}-\left(1-\alpha \right)\left(2c_{n}+c_{t}e_{1}+c_{t}e_{2}\right)\right]}{2[2C\left(2-\alpha \right)-\left(1-\alpha \right){\left(\Delta -b\right)}^{2}]}$
$C_{s}^{NG*}$	$\frac{C^{2}{\left[a_{1}-a_{2}-\left(1+\alpha \right)c_{t}\left(e_{1}-e_{2}\right)\right]}^{2}}{4{\left[2C\left(2+\alpha \right)-\left(1+\alpha \right){\left(\Delta -b\right)}^{2}\right]}^{2}}+\frac{C^{2}{\left[a_{1}+a_{2}-\left(1-\alpha \right)\left(2c_{n}+c_{t}e_{1}+c_{t}e_{2}\right)\right]}^{2}}{4{\left[2C\left(2-\alpha \right)-\left(1-\alpha \right){\left(\Delta -b\right)}^{2}\right]}^{2}}$
$\lambda E^{NG*}$	$\frac{\lambda \left(e_{1}-e_{2}\right)C\left[a_{1}-a_{2}-\left(1+\alpha \right)c_{t}\left(e_{1}-e_{2}\right)\right]}{2[2C\left(2+\alpha \right)-\left(1+\alpha \right){\left(\Delta -b\right)}^{2}]}+\frac{\lambda \left(e_{1}+e_{2}\right)C\left[a_{1}+a_{2}-\left(1-\alpha \right)\left(2c_{n}+c_{t}e_{1}+c_{t}e_{2}\right)\right]}{2[2C\left(2-\alpha \right)-\left(1-\alpha \right){\left(\Delta -b\right)}^{2}]}$
$W^{NG*}$	$\Pi_{m}^{NG*}+\Pi_{r}^{NG*}+\Pi_{G}^{NG*}+C_{s}^{NG*}-\lambda E^{NG*}$

**Table 4 ijerph-19-09244-t004:** Main results for the decentralized model of retailer with green marketing, *i* = 1, 2.

Notations	Values
$s_{i}^{HG*}$	$\frac{\beta C[a_{1}+a_{2}-\left(1-\alpha \right)\left(2c_{n}+c_{t}e_{1}+c_{t}e_{2}\right)]}{\left(1-\alpha \right)\eta \left[2C\left(2-\alpha \right)-\left(1-\alpha \right){\left(\Delta -b\right)}^{2}\right]-2\beta^{2}C}$
$p_{i}^{HG*}$	$\frac{C\left(1-\alpha^{2}\right)\left[a_{3-i}-c_{t}\left(e_{3-i}-e_{i}\right)\right]+\left(a_{i}+\alpha a_{3-i}\right)\left[7C+5\alpha C-2\left(1+\alpha \right){\left(\Delta -b\right)}^{2}\right]}{2\left(1-\alpha^{2}\right)\left[2C\left(2+\alpha \right)-\left(1+\alpha \right){\left(\Delta -b\right)}^{2}\right]}-\frac{\left[\left(1-\alpha \right)\eta -2\beta^{2}\right]s^{HG*}}{2(1-\alpha )\beta }$
$w_{i}^{HG*}$	$\frac{8c_{n}C+\left[2C-{\left(\Delta -b\right)}^{2}\right]\left[a_{i}-a_{3-i}+\left(1+\alpha \right)\left(4c_{n}-c_{t}e_{i}+c_{t}e_{3-i}\right)\right]}{4[2C\left(2+\alpha \right)-\left(1+\alpha \right){\left(\Delta -b\right)}^{2}]}+\frac{\left[2C-{\left(\Delta -b\right)}^{2}\right]\left(1-\alpha \right)\eta s^{HG*}}{4\beta C}$
$\tau_{i}^{HG*}$	$\frac{\left(\Delta -b\right)[a_{i}-a_{3-i}-\left(1+\alpha \right)c_{t}\left(e_{i}-e_{3-i}\right)]}{4[2C\left(2+\alpha \right)-\left(1+\alpha \right){\left(\Delta -b\right)}^{2}]}+\frac{\left(\Delta -b\right)\left(1-\alpha \right)\eta s^{HG*}}{4\beta C}$
$\Pi_{m_{i}}^{HG*}$	$\frac{4C-{\left(\Delta -b\right)}^{2}}{4C}{\left(\frac{C[a_{i}-a_{3-i}-\left(1+\alpha \right)c_{t}\left(e_{i}-e_{3-i}\right)]}{2[2C\left(2+\alpha \right)-\left(1+\alpha \right){\left(\Delta -b\right)}^{2}]}+\frac{\left(1-\alpha \right)\eta s^{HG*}}{2\beta }\right)}^{2}$
$\Pi_{r}^{HG*}$	$\frac{C{\left[a_{1}-a_{2}-\left(1+\alpha \right)c_{t}\left(e_{1}-e_{2}\right)\right]}^{2}}{4\left(1+\alpha \right)[2C\left(2+\alpha \right)-\left(1+\alpha \right){\left(\Delta -b\right)}^{2}]}+\frac{\left[a_{1}+a_{2}-\left(1-\alpha \right)\left(2c_{n}+c_{t}e_{1}+c_{t}e_{2}\right)\right]\eta s^{HG*}}{4\beta }$
$E^{HG*}$	$\frac{\left(e_{1}-e_{2}\right)C\left[a_{1}-a_{2}-\left(1+\alpha \right)c_{t}\left(e_{1}-e_{2}\right)\right]}{2[2C\left(2+\alpha \right)-\left(1+\alpha \right){\left(\Delta -b\right)}^{2}]}+\frac{\left(e_{1}+e_{2}\right)\left(1-\alpha \right)\eta s^{HG*}}{2\beta }$
$\Pi_{G}^{HG*}$	$\frac{c_{t}\left(e_{1}-e_{2}\right)C\left[a_{1}-a_{2}-\left(1+\alpha \right)c_{t}\left(e_{1}-e_{2}\right)\right]}{2[2C\left(2+\alpha \right)-\left(1+\alpha \right){\left(\Delta -b\right)}^{2}]}+\frac{c_{t}\left(e_{1}+e_{2}\right)\left(1-\alpha \right)\eta s^{HG*}}{2\beta }$
$C_{s}^{HG*}$	$\frac{C^{2}{\left[a_{1}-a_{2}-\left(1+\alpha \right)c_{t}\left(e_{1}-e_{2}\right)\right]}^{2}}{4{\left[2C\left(2+\alpha \right)-\left(1+\alpha \right){\left(\Delta -b\right)}^{2}\right]}^{2}}+\frac{{\left(1-\alpha \right)}^{2}\eta^{2}{\left(s^{HG*}\right)}^{2}}{4\beta^{2}}$
$\lambda E^{HG*}$	$\frac{\lambda \left(e_{1}-e_{2}\right)C\left[a_{1}-a_{2}-\left(1+\alpha \right)c_{t}\left(e_{1}-e_{2}\right)\right]}{2[2C\left(2+\alpha \right)-\left(1+\alpha \right){\left(\Delta -b\right)}^{2}]}+\frac{\lambda \left(e_{1}+e_{2}\right)\left(1-\alpha \right)\eta s^{HG*}}{2\beta }$
$W^{HG*}$	$\Pi_{m}^{HG*}+\Pi_{r}^{HG*}+\Pi_{G}^{HG*}+C_{s}^{HG*}-\lambda E^{HG*}$

**Table 5 ijerph-19-09244-t005:** Main results for the partial cooperation model of retailer with green marketing, *i* = 1, 2.

Notations	Values
$s_{i}^{PC*}$	$\frac{\beta C[a_{1}+a_{2}-\left(1-\alpha \right)\left(2c_{n}+c_{t}e_{1}+c_{t}e_{2}\right)]}{\left(1-\alpha \right)\eta \left[4C-\left(1-\alpha \right){\left(\Delta -b\right)}^{2}\right]-2\beta^{2}C}$
$p_{i}^{PC*}$	$\frac{C\left(1-\alpha^{2}\right)\left[a_{3-i}-c_{t}\left(e_{3-i}-e_{i}\right)\right]+\left(a_{i}+\alpha a_{3-i}\right)\left[7C+\alpha C-2\left(1+\alpha \right){\left(\Delta -b\right)}^{2}\right]}{2\left(1-\alpha^{2}\right)\left[4C-\left(1+\alpha \right){\left(\Delta -b\right)}^{2}\right]}-\frac{\left[\left(1-\alpha \right)\eta -2\beta^{2}\right]s^{PC*}}{2(1-\alpha )\beta }$
$w_{i}^{PC*}$	$\frac{8c_{n}C\left(1+\alpha \right)+\left[2C-\left(1+\alpha \right){\left(\Delta -b\right)}^{2}\right]\left[a_{i}-a_{3-i}+\left(1+\alpha \right)\left(4c_{n}-c_{t}e_{i}+c_{t}e_{3-i}\right)\right]}{4\left(1+\alpha \right)[4C-\left(1+\alpha \right){\left(\Delta -b\right)}^{2}]}+\frac{\left[2C-\left(1-\alpha \right){\left(\Delta -b\right)}^{2}\right]\eta s^{PC*}}{4\beta C}$
$\tau_{i}^{PC*}$	$\frac{\left(\Delta -b\right)[a_{i}-a_{3-i}-\left(1+\alpha \right)c_{t}\left(e_{i}-e_{3-i}\right)]}{4[4C-\left(1+\alpha \right){\left(\Delta -b\right)}^{2}]}+\frac{\left(\Delta -b\right)\left(1-\alpha \right)\eta s^{PC*}}{4\beta C}$
$\Pi_{m}^{PC*}$	$\frac{{\left[a_{1}-a_{2}-\left(1+\alpha \right)c_{t}\left(e_{1}-e_{2}\right)\right]}^{2}C}{8\left(1+\alpha \right)[4C-\left(1+\alpha \right){\left(\Delta -b\right)}^{2}]}+\frac{\left[4C-\left(1-\alpha \right){\left(\Delta -b\right)}^{2}\right]\left(1-\alpha \right)\eta^{2}{\left(s^{PC*}\right)}^{2}}{8\beta^{2}C}$
$\Pi_{r}^{PC*}$	$\frac{{\left[a_{1}-a_{2}-\left(1+\alpha \right)c_{t}\left(e_{1}-e_{2}\right)\right]}^{2}C}{4\left(1+\alpha \right)[4C-\left(1+\alpha \right){\left(\Delta -b\right)}^{2}]}+\frac{\left[a_{1}+a_{2}-\left(1-\alpha \right)\left(2c_{n}+c_{t}e_{1}+c_{t}e_{2}\right)\right]\eta s^{PC*}}{4\beta }$
$E^{PC*}$	$\frac{\left(e_{1}-e_{2}\right)C\left[a_{1}-a_{2}-\left(1+\alpha \right)c_{t}\left(e_{1}-e_{2}\right)\right]}{2[4C-\left(1+\alpha \right){\left(\Delta -b\right)}^{2}]}+\frac{\left(e_{1}+e_{2}\right)\left(1-\alpha \right)\eta s^{PC*}}{2\beta }$
$\Pi_{G}^{PC*}$	$\frac{c_{t}\left(e_{1}-e_{2}\right)C\left[a_{1}-a_{2}-\left(1+\alpha \right)c_{t}\left(e_{1}-e_{2}\right)\right]}{2[4C-\left(1+\alpha \right){\left(\Delta -b\right)}^{2}]}+\frac{c_{t}\left(e_{1}+e_{2}\right)\left(1-\alpha \right)\eta s^{PC*}}{2\beta }$
$C_{s}^{PC*}$	$\frac{C^{2}{\left[a_{1}-a_{2}-\left(1+\alpha \right)c_{t}\left(e_{1}-e_{2}\right)\right]}^{2}}{4{\left[4C-\left(1+\alpha \right){\left(\Delta -b\right)}^{2}\right]}^{2}}+\frac{{\left(1-\alpha \right)}^{2}\eta^{2}{\left(s^{PC*}\right)}^{2}}{4\beta^{2}}$
$\lambda E^{PC*}$	$\frac{\lambda \left(e_{1}-e_{2}\right)C\left[a_{1}-a_{2}-\left(1+\alpha \right)c_{t}\left(e_{1}-e_{2}\right)\right]}{2[4C-\left(1+\alpha \right){\left(\Delta -b\right)}^{2}]}+\frac{\lambda \left(e_{1}+e_{2}\right)\left(1-\alpha \right)\eta s^{PC*}}{2\beta }$
$W^{PC*}$	$\Pi_{m}^{PC*}+\Pi_{r}^{PC*}+\Pi_{G}^{PC*}+C_{s}^{PC*}-\lambda E^{PC*}$

**Table 6 ijerph-19-09244-t006:** Comparison of four models.

Model	($w_{1}$; $\tau_{1}$)	($w_{2}$; $\tau_{2}$)	(*s*; $p_{1}$; $p_{2}$)	$\Pi_{m_{1}}$	$\Pi_{m_{2}}$	$\Pi_{m}$	$\Pi_{r}$	Total Profit	Social Welfare
NG	(58.23; 0.20)	(42.88; 0.12)	(-; 178.61; 129.50)	1578.64	565.64	2144.27	5145.08	7289.35	4223.48
HG	(60.90; 0.22)	(45.55; 0.13)	(6.25; 187.98; 138.86)	1807.10	705.49	2512.59	5575.52	8088.11	4856.64
PC	(67.03; 0.19)	(53.61; 0.10)	(5.22; 193.01; 145.82)	-	-	2530.41	4733.55	7263.96	4390.77

## Data Availability

Not applicable.

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
