# Peer review of "Optimal Decisions in a Multi-Party Closed-Loop Supply Chain Considering Green Marketing and Carbon Tax Policy"

_ijerph, 2022, doi:10.3390/ijerph19159244_

Round 1
Reviewer 1 Report
This research tries to consider a closed-loop supply chain consisting of two competing manufacturers and a dominant retailer, where the two manufacturers collect used products directly from their customers. Although I think the research can include some potential for publication, this version suffers from some major issues as follows:
The abstract should contain the following sections:
1) What is the general problem addressed in the paper?
2) What is the specific research question to be answered?
3) What are the means and methods used by the authors to answer the stated question?
4) What is the answer to the research question?
5) Why is the answer important and for whom specifically?
2 The introduction section is a very important part of the manuscript that should justify the necessity of the research based on more recent more relevant papers. However, the introduction provided by the researchers is not enough for such research and such a well-known journal. The introduction should be provided from more general to detail including problem definition, research gaps, questions, novelties, and contributions to the literature. Some other more recent more relevant papers provide a good modeling approach in dealing with the CLSC, optimal approaches, and simulation ones. Is the system decentralized or centralized? I see no declaration about this, while it is important. You should claim these characteristics based on some research works that work on this feature. Accordingly, I suggest completely reorganizing the introduction section. The research is more relevant to the concept of sustainable closed loop supply chain. Why the authors do not do their best to claim this concept and the modeling approaches that are provided to formulate it. I suggest the authors consider more recent references from 2021 and 2022, otherwise, it can not be justified that the authors walk on the frontier of the research.
Based on the above comment the authors are suggested to reorganize the literature review. They should focus on the modeling approaches that are used to model the CLSCs and the sustainable CLSCs even by using a summary table focusing on the novelties and contributions of the research at hand.
Sections 3 and 4 should be reorganized. It is provided in a bulky way. The research method is not clear. A good flowchart including problem formulation, reformulation, and solution can be helpful. The authors are suggested to provide these components as subsections of the section.
The results make no sense. The authors should further extend the results and discussions. Where are the managerial policy suggestions resulting from the research?
Reviewer 2 Report
Please see the attached file.
